# Does a proxy measure up?: A framework to assess and convey proxy reliability

F. Garrett Boudinot[1], Joseph Wilson[2]

[1] Department of Geological Sciences and Institute of Arctic and Alpine Research, University of Colorado Boulder,

5    Boulder, Colorado, 80309, USA

[2] Department of Philosophy and Center for the Study of Origins, University of Colorado Boulder, Boulder, Colorado, 80309, USA

*Correspondence to:* F. Garrett Boudinot (garrett.boudinot@colorado.edu)

**Abstract.** Earth scientists describe a wide range of observational measurements as "proxy measurements." By referring to such a vast body of measurements simply as "proxy," workers dilute significant differences in the various ways that measurements relate to the phenomena they intend to describe. The limited language around these measurements makes it difficult for the non-specialist to assess the reliability and uncertainty of data generated from "proxy" measurements. Producers and reviewers of proxy data need a common framework for conveying proxy measurement methodology, uncertainty, and applicability for a given study.

We develop a functional distinction between different forms of measurement based on the different ways that their outputs (values, interpretations) relate to the phenomena they intend to describe (e.g., temperature). Paleotemperature measurements, which intend to represent the temperature of systems in Earth's ancient past, are used as a case study to examine and apply this new functional proxy definition. We explore the historical development and application of two popular paleotemperature proxies, calcite $\delta^{18}O$ and $TEX_{86}$, to illustrate how different measurements relate to the phenomena they intend to describe. Both proxies are vulnerable to causal factors that interfere with their relationship with temperature ("confounding causal factors"), but address those interfering causal factors in different ways. While the goal of proxy development is to fully identify, quantify, and calibrate to all confounding causal factors, the reality of proxy applications, especially for past systems, engenders unavoidable and potentially significant uncertainties. We propose a framework that allows researchers to be explicit about the limitations of their proxies, and identify steps for further development. This paper underscores the ongoing effort and continued need for critical examination of proxies throughout their development and application, particularly in Earth history, for reliable proxy interpretation.

## 1 Introduction

Proxy measurements are used to provide information about otherwise elusive properties of systems in Earth's past, present, and worlds beyond. With a growing interest in quantitatively measuring these properties more precisely and in new environments, the diversity of proxies has increased dramatically. While "proxy" is often used to differentiate indirect (e.g., geochemical, physical, etc.) measurements from more "direct" forms of observational measurement, neither of these terms provides insight into the reliability or applicability of different measurements. Even "direct" forms of measurement can be considered proxy in this sense; all involve some level of observational "indirectness" (Wilson and Boudinot, *in review*). Earth scientists are particularly aware of the nuances of measurement applicability – as workers look farther back in time, the reliability of a measurement (i.e., our understanding of what that measurement represents) typically becomes less certain. A standardized framework for conveying how proxy measurements relate to different systems and phenomena would be widely useful for describing these complex associations to non-specialists, students, modelers, and other proxy users.

The goal of this paper is to describe how methods of observational measurements differ in the ways their outputs (values, data, interpretations) relate to the phenomena they intend to describe. All forms of observational measurement are influenced by factors that are not the property being measured. We provide insight into the assumptions behind the interpretation and development of different forms of measurement, with the goal of more clearly describing those assumptions and uncertainties in the context of data interpretations.

We use paleotemperature measurements, which intend to represent changes in the temperature of systems in Earth's ancient past, as a case study given the growing interest in paleotemperatures (Fig. 1), the diversity of measurements available, and the field's relationship to unknown changes in the Earth-climate system through time. We propose a theoretical framework and language that can more accurately distinguish different measurement-property relationships, which we hope will lead to more robust measurement calibrations, more transparent measurement outputs, and stronger interpretations. While paleoclimate is the focus below, the ideas described here apply to observational measurements across many fields of science.

## 2 Functional distinctions for proxy measurements

The placement of measurements in two overarching groups, proxy and direct, is particularly common in climate sciences (NOAA National Centers for Environmental Information; Jansen et al., 2007). Philosophical work (Wilson and Boudinot, *in review*) has pointed out the need for clarification behind the definition of proxy measurements as "indirect" and non-proxy measurements as "direct," and questioned how proxies can provide reliable measurements in spite of such perceived indirectness. While many have referred to oxygen isotopes in calcite ($\delta^{18}O_{calcite}$) as a proxy- and the mercury thermometer as a direct-measurement (NOAA National Centers for Environmental Information; Jansen et al., 2007), both scientists and philosophers of science have pointed out a lack of difference in observational "directness" between the two measurement techniques (e.g., Ruddiman, 2008; Wilson and Boudinot, *in review*). The mercury thermometer measures temperature via the observable thermal expansion of mercury as a function of temperature, while the $\delta^{18}O_{calcite}$ measures paleotemperature via observable variation of $^{18}O$ incorporation into calcite ($CaCO_3$) as a function of temperature, resulting from the differences in vibrational energies of different oxygen isotopes (i.e., $^{16}O$, $^{17}O$, $^{18}O$). In other words, neither produces a "direct" measurement of temperature; both rely on the observation of some effect of temperature in a system.

Each of these measurements are also influenced by other non-temperature causal factors. Mercury expansion is not only a function of temperature, but also of the partial pressure of the atmosphere and expansion dynamics of liquid mercury. Similarly, $\delta^{18}O_{calcite}$ is influenced by the $\delta^{18}O$ of the surrounding water ($\delta^{18}O_{H2O}$; Urey, 1948), the pH of the surrounding water (Spero et al., 1997), and if biomineralized by calcifying organisms, biological kinetic effects on $^{18}O$ incorporation (Bemis et al., 1998; Ravelo and Hillaire-Marcel, 2007). Philosophers attuned to the conceptual and epistemic issues regarding different forms of scientific measurement (e.g. Suppes, 1951; Franklin, 1990; Chang, 2004; Van Fraassen, 2010) have recently proposed that proxies differ from other forms of measurement in how they account for these *confounding causal factors* (CCFs; see glossary of terms).

Under this definition, non-proxy measurements are those that have been manufactured/designed to eliminate all of the potential effects of known CCFs on the measurement output. Because these non-proxy measures control which parts of the system contribute to the final measurement outputs, we refer to them as *controlled* measurements (see

glossary of terms). Mercury thermometers, for example, are manufactured with a glass casing that controls the atmospheric pressure within the thermometer. The glass case eliminates variation in non-temperature CCFs (e.g., changes in atmospheric pressure, potential for fluid exchange) such that the measured signal can only represent the phenomena in question, temperature. The lines on the thermometer are calibrated to the thermodynamic properties of mercury, such that a specific volumetric expansion of mercury is a causal result of the specific local temperature. In this way, the mercury thermometer is used to perform a controlled measurement.

While the process is more sophisticated, the digital thermometers more commonly used today also control all known CCFs within the instrument to provide a single, calibrated temperature value. For those digital thermometers that use electrical resistance, for example, the built-in computer immediately converts an electrical resistance reading to temperature, and is calibrated to reduce the influence of non-temperature effects on such resistance, including the composition, length, and width of the metal probe used in the thermometer. Because the CCFs that influence the relationship between electrical resistance and temperature are accounted for and calibrated to immediately in the act of measurement, digital thermometers, too, are used to perform controlled measurements.

Proxy measurements are distinct because their process of measurement does not rule out all CCFs (see glossary of terms). This means that the original signal from the analytical measurement must be subject to further manipulation, such as incorporation into a calibration. Those calibrations are based on the field's best understanding of the drivers of that measured property, and quantitatively attempt to minimize the influence of CCFs to produce a value that represents the phenomena in question (Fig. 2). For example, $\delta^{18}O_{calcite}$ is a proxy measurement because $\delta^{18}O_{calcite}$ is measured simply as a ratio of $^{18}O$ to $^{16}O$ of a calcite sample compared to an isotopic standard, and alone that analytical measurement does not reflect temperature. To measure temperature using $\delta^{18}O_{calcite}$, paleoclimate researchers must incorporate into a calibration information about other parts of the system that influence the inclusion of $^{18}O$ into calcite, such as the $\delta^{18}O_{H2O}$ of the surrounding water, and any potential biological effects of calcification. Because most proxy applications do not allow the researcher to produce controlled measurements of each of those CCFs, the output from a proxy is at best an "estimate" (i.e., the $\delta^{18}O_{calcite}$ proxy measurement produces paleotemperature estimates).

The term "indicator" is often used synonymously with "proxy" or even "measurement" (e.g., *"Application of the Ce anomaly as a paleoredox indicator,"* German and Elderfield, 1990;

"Using fossil leaves as paleoprecipitation indicators,' Wilf et al., 1998; "Stomatal density and stomatal index as indicators of paleoatmospheric $CO_2$ concentrations," Royer, 2001; "indicator of relative changes in sea surface temperature," Hollis et al., 2019; "Palaeoecological proxies…include crustacean Ostracoda…their indicator species…are sensitive to deoxygenation and eutrophication," Yasuhara et al., 2019). The use of this term for such wide range of

applications highlights the lack of clarity in the existing literature, which eventually leads to a lack of clarity in the dissemination of resulting information. While all measurements do "indicate" the quality of some property, they do so in different ways, and are accompanied by quite different levels of reliability and uncertainty. The proposed distinction between proxy and controlled measurements, and within proxy measurements (see below), is aimed to provide a

more clear distinction to the discussion of measurements and their outputs – and CCFs provide such distinction.

The importance of CCFs for proxy measurements was recognized in the development of the first quantitative paleotemperature proxy, $\delta^{18}O_{calcite}$. Harold Urey first described the thermodynamic relationship between $\delta^{18}O_{calcite}$ and calcite formation temperatures through a

simple linear calibration that relates $\delta^{18}O_{calcite}$ to temperature in degrees Celsius (Urey 1948). Urey discussed two important CCFs influencing the $\delta^{18}O_{calcite}$ relationship with temperature that could have changed significantly through geologic time and space: $\delta^{18}O_{H2O}$ of the (mean) global ocean, and $\delta^{18}O_{H2O}$ of local waters surrounding the precipitating carbonate. While the early reports posited global $\delta^{18}O_{H2O}$ changes on long timescales (millions of years) were a result of

rock weathering, later work showed that global $\delta^{18}O_{H2O}$ had varied significantly on much shorter timescales (tens of thousands of years) due to fluctuations of global ice volume (Emiliani, 1955). The uncertainty of mean ocean $\delta^{18}O_{H2O}$ becomes greater for older periods of Earth history, due to currently unconstrained conditions such as ancient ocean latitudinal gradient effects (i.e., reduced latitudinal temperature gradient and resultant local $\delta^{18}O_{H2O}$ ~100 million years ago) and silicate

weathering rates (Urey et al., 1951). Such temporal variations in baseline characteristics of Earth systems contributes to a widespread positive relationship between geologic time and uncertainty, making all CCFs more difficult to constrain in increasing geologic age. As such, different

temporal applications of a single proxy can dramatically change that proxy estimate's uncertainty.

The potential for unknown CCFs exists even for well-calibrated proxy systems and control measurements (Wilson and Boudinot, *in review*). While the mercury thermometer successfully controls for its relevant CCFs, a hypothetical application that reveals a theretofore unknown CCF would lead us to no longer consider the thermometer a controlled measurement, at least until it were manufactured in a way to also remove the effects of that CCF. The potential

for the existence of unknown CCFs necessitates cautious interpretations of all measurements, particularly those in development or under new applications. But how exactly are CCFs incorporated into proxies?

**3 Assessing a proxy**

**3.1 Situating proxies on a spectrum**

CCFs are incorporated into proxy measurements through a calibration equation (Fig. 2), which provides a quantitative representation of the relative influence of each causal factor that contributes to the measured property. Using the calibration, researchers can effectively remove the influence of CCFs, and produce an estimate of the phenomenon in question. However, the

extent to which calibrations identify and address CCFs differs greatly between different proxies.

        We place proxy measurements along a spectrum that can illustrate the diversity of how proxies relate to CCFs (Fig. 3a). Controlled measurements, with all CCFs known and controlled for (e.g., mercury thermometer), occupy one end of the spectrum. On the other end of the spectrum are proxy measures that have yet to be calibrated in a way that accounts for their CCFs,

such that only a correlation is proposed (*correlation-constrained proxy;* see glossary of terms), carrying high uncertainty in what CCFs there are and/or their precise causal influence. In between the two ends of the spectrum are proxies which have a calibration that accounts for the CCFs' influence on the measurement output, and are accompanied by a quantitative measurement (*observation-constrained proxy*) or quantitative inference (*inference-constrained*

*proxy*) of those CCFs (Fig. 3a; see glossary of terms). By situating any measurement along this spectrum, one can assess how much the measured value is affected by something other than the property in question (i.e., the *potential uncertainty*, Fig. 3b, see below) such as $\delta^{18}O_{H2O}$ instead of temperature.

Controlled measurements work the same across locations and through time. A mercury
thermometer should have the same level of accuracy and precision in a high-altitude, low-humidity study site as in a low-altitude, high-humidity site. In an ideal situation, all proxy measurements would be developed in a way that they could be controlled measurements. Unfortunately, and particularly in paleo applications, the certainty ascribed to the mercury expansion calibration is not easily attainable or validated. Furthermore, even controlled measurements can be complicated by work in "extreme" environments, where temperatures may exceed the minimum or maximum range to which the thermometer is calibrated (e.g., beyond the boiling point of mercury). Thus, how a measurement's calibration is developed and utilized determines the situations and uncertainty for that measurement's application.

To illustrate the proxy range of the spectrum, we situate $\delta^{18}O_{calcite}$ as either an *observation-constrained proxy* or an *inference-constrained proxy* depending on how CCFs are quantitatively accounted for (Fig. 3a). When the $\delta^{18}O_{H2O}$ value in the temperature calibration derives from an independent measurement (proxy or controlled) of the $\delta^{18}O_{H2O}$ of the water from which the calcite precipitated, then the proxy is an *observation-constrained proxy;* values to account for the CCFs in the calibration are accounted for with empirical observations (Fig. 3a). These components of the calibration can be accounted for with information from proxy or controlled measurements, with the latter contributing less uncertainty given the constraints on CCFs in controlled measurements (see below).

On the other hand, in instances where $\delta^{18}O_{H2O}$ cannot be estimated from a proxy measurement, such as in deeper-time applications, the researcher must provide an *inference* (i.e., reasoned approximation) of local $\delta^{18}O_{H2O}$. Based on the extrapolation of a well-known system to a lesser-known system, *inference-constrained proxy* measurements inherently present a more biased estimate, due to biases in the researchers' inference of that system, rather than empirical evidence (Fig. 3b). For example, some workers have inferred $\delta^{18}O_{H2O}$ values for the $\delta^{18}O_{calcite}$ paleotempearture calibration by applying a first-order estimate of $\delta^{18}O_{H2O}$ based on certain characteristics of the system in question, such as a mean $\delta^{18}O_{H2O}$ value that applies to any "non-glacial world" (O'Brien et al., 2017). This mean value was modified to represent $\delta^{18}O_{H2O}$ of local waters (where calcite was precipitated) by adjusting the mean $\delta^{18}O_{H2O}$ based on modern latitudinal $\delta^{18}O_{H2O}$ variability (e.g., O'Brien et al., 2017). This inference is still based on quantitative measurements (e.g., modern $\delta^{18}O_{H2O}$ latitudinal trends), but requires several

inferences that assume that two systems are similar (all ice free oceans in Earth history are isotopically similar; latitudinal $\delta^{18}O_{H2O}$ variability is similar between the past and present).

Because different CCFs can be more or less easily accounted for, many calibrations require a combination of inference and observation to produce a final estimate of the target property. In other words, many proxy applications use both observation- and inference-constraints to satisfy a calibration. Because that estimation is accompanied by uncertainty that is not easily quantifiable (e.g., uncertainty associated with assumptions made by the researcher, rather than analytical uncertainty, see below), the potential uncertainty for inference-constrained proxies is larger than those that are observation-constrained.

Moving further away from controlled measurements on our spectrum, we find proxy measurements that are *correlated* with temperature, but the CCFs are not fully or quantitatively accounted for in a calibration; here, the CCFs are unknown (or roughly understood), though a corollary relationship is identified. It is functionally impossible to accurately assess the uncertainty of estimates produced by these measurements (Fig. 3b), as the causal factors influencing the measurement are simply unknown or not quantitatively represented in a calibration. The signal from such *correlation-constrained proxy* could be entirely driven by some CCF, but would interpreted as driven by the property intended to be measured.

An example of a correlation-constrained proxy is the present incarnation of the TEX$_{86}$ paleotemperature proxy. In 2002, workers identified a suite of sedimentary hydrocarbons that shared a similar structure, but contained a different number of cyclic moieties (Schouten et al., 2002; Fig. 3). Relative abundances of these isoprenoidal glycerol diether glycerol tetraether (isoGDGT) compounds with different cyclic moieties were represented by a ratio (Table 1). When these compounds were recovered from modern sediments and this ratio was calculated, a clear correlation with the surface water temperature at the sample location was identified. In other words, the number of cyclic moieties in the sedimentary isoGDGTs were correlated with the surface water temperatures at the location that they were found. Using statistical (regression) analyses of a suite of modern sediments and sea surface temperature measurements, a calibration was produced, and the authors proposed this molecular ratio as a quantitative paleotemperature proxy (Schouten et al., 2002). A physiological response was posited to explain the relationship – less cyclic moieties contributed to a more malleable lipid membrane, which would be advantageous in cooler waters.

In the ensuing years, several revelations about these molecules came to light: they seemed to be produced predominantly by Thaumarcheota, a type of marine archaea that live well below the sea surface. Additionally, field and culture calibrations from variable environments produced different calibrations (i.e., different slopes and y-intercepts to describe the correlation between the isoGDGT ratio and temperature; Table 1) and even different ratios (e.g., $TEX_{86}^L$ for low-temperature regions; Table 1). If the ratio of isoGDGT cyclicity directly represented temperature, then why would that ratio be different depending on the study design, location, and time period? And if the calibration accurately accounted for the CCFs contributing to the effect of temperature on isoGDGT cyclicity, why would it be different from place to place?

These questions are driving fundamental research in understanding the *mechanistic relationships* between $TEX_{86}$ and temperature. Several important advances in this mechanistic understanding have already been produced: culture and field experiments demonstrated that the cyclic moieties represented a metabolic response to energy demands, growth phase, nutrient availability, and ecosystem composition, rather than a physiological response to temperature (Elling et al., 2014; Qin et al., 2015; Hurley et al., 2016; Polik et al., 2018). These studies advance $TEX_{86}$ beyond the corollary relationship (i.e., colder temperatures makes more cyclic moieties) into a nuanced, yet more accurately representative, understanding of all causal factors and their mechanisms (i.e., relationship between sea surface temperatures and ammonia and oxygen availability, which impacts archaeal metabolic energy demands). However, while work on $TEX_{86}$ drivers suggest that non-temperature factors cause variations in isoGDGT cyclization, $TEX_{86}$ application studies continue to report a specific temperature value. The argument behind continued $TEX_{86}$ applications is the correlation of ammonia oxidation rates and temperature in *most modern settings* (Hurley et al., 2016), while many studies have suggested that ammonia or oxygen concentrations in past environments likely varied in a way that did not correlate with temperature (e.g., Liu et al., 2009; Polik et al., 2018). This proxy's CCFs need full consideration in experimental design and interpretation for it to be truly quantitative – and its uncertainty appropriately reported.

**3.2 Discussing proxy data**

A clear distinction should be made between various forms and degrees of uncertainty related to proxy measurements (see glossary of terms). All proxy measurements are the result of some

analysis (e.g., $\delta^{18}O_{calcite}$ as the normalized ratio of $^{18}O$ to $^{16}O$ in a sample) and incorporation into a calibration (e.g., $\delta^{18}O_{calcite}$ as a function of temperature, $\delta^{18}O_{H2O}$, and biological effects; Fig. 1), from which derives three forms of uncertainty. The first is *analytical uncertainty*, which is

simply the uncertainty associated with the precision and accuracy of the analytical instrument. For oxygen isotopes in calcite, this would include the isotope ratio mass spectrometer's precision and accuracy when determining the ratio of $^{18}O$ to $^{16}O$ of a sample normalized to a standard. We argue that analytical uncertainty can always be quantified using standards, and is unique from the unquantifiable uncertainties that could arise from sample processing, human error, etc. Those

unquantifiable uncertainties in analysis are grouped with other unquantifiable uncertainties associated in a calibration, such as unknown CCFs, in what we call *potential uncertainties* (Fig. 3b). The distinction between factors that fall in the potential uncertainty group versus the analytical uncertainty group is defined by quantitation. Errors from sample processing that might introduce uncertainty can be quantified using standards throughout processing steps to measure

sample losses, for example. Incorporation of that measured processing error into the analytical uncertainty would reduce the potential uncertainty and more accurately reflect that analytical uncertainty. For example, hydrocarbon standards might be incorporated into a sedimentary sample before hydrocarbon extraction, such that the researcher can quantify if any hydrocarbons, including isoGDGTs, are lost or altered throughout the in-lab processing. Researchers could

report or normalize to that loss and alteration, more transparently reflecting the uncertainty in the analysis. However, some potential uncertainties will always exist in a non-quantifiable manner, such as unknown CCFs or un-measureable changes in CCFs through time. Because the error in an inference-constrained proxy might not be quantifiable (i.e., logical deductions might not have a quantifiable uncertainty), its potential uncertainty will always be higher than an observation-

constrained proxy, in which the analytical uncertainty in that estimate used in the calibration can be quantified (red and blue lines in Fig. 3b).

      The final type of uncertainty is the *reported uncertainty*, which should ideally cover (either quantitatively or in discussion) both analytical and potential uncertainties (Fig. 3b). However, for many proxies, the reported uncertainty varies widely in practice. For example, the

variety of isoGDGT ratios and calibrations (Table 1), and the lack of codified reporting standards used in the expression of TEX$_{86}$-derived paleotemperatures, leads to notable variability in the reported uncertainty associated with TEX$_{86}$, particularly between different groups of researchers

(blue bars in Fig. 3b). Some researchers reporting $TEX_{86}$-derived paleotemperature estimates, for example, plot no error bars and report in-text the analytical uncertainty from the calibration used

and replicate analyses (e.g., Woelders et al., 2017), or provide no analytical uncertainty (e.g., Slujis et al., 2006). Others have included only the analytical uncertainty derived from the calibration used (e.g., Hollis et al., 2012; Ho et al., 2014). Some reporting has shown analytical uncertainties from replicate analyses combined with the analytical uncertainties of calibration statistics as error windows on plots, but have not discussed in detail other potential uncertainties,

such as changes in the known (but not calibrated-to) CCFs (e.g., Tierney et al., 2010). Others have plotted the analytical uncertainty from replicate analyses as error bars/windows on a plot, and discussed further potential uncertainties in text, which we find provides a more complete reported uncertainty (e.g., Shevenell et al., 2011). Because potential uncertainty is by-definition unquantifiable, it might not be incorporated into quantitative data presentation styles such as

Cartesian plots, but can certainly be discussed in light of the existing work on $TEX_{86}$ CCFs.

 Importantly, researchers have already taken important steps to communicate the reliability of proxy data relative to other measurements in reviews, conference sessions, and proxy assessment compilations (e.g. Ravelo and Hillaire-Marcel, 2007; Newman et al., 2016; Hollis et al., 2019; Wilson and Boudinot, 2019). For example, the Paleoclimate Modelling

Intercomparison Project (PMIP)'s appraisal of proxy data for the Intergovernmental Panel on Climate Change (IPCC) reports (Hollis et al., 2019) provides an in-depth description of the paleotemperature proxies used to inform the IPCC reports. The appraisal describes each proxy's theoretical background, which gives data generators and modelers a better understanding of the biogeochemical processes that relate each proxy to temperature. The assessment then describes

strengths and weaknesses of each proxy relative to the other measurements, which can guide users in determining which proxy may be best suited for a given study, as well as providing considerations for the interpretation of the resulting data. Finally, the assessment provides "recommended methodologies," which includes analytical recommendations, a single recommended calibration, and other best-practices for reporting proxy data and interpretations.

By providing a consensus presentation of recommended methodologies particularly, the PMIP proxy assessment and similar projects constitute an important means for standardizing data assessment and reporting, and guiding proxy users in developing study designs. The framework

presented here will improve those methods by providing direct language (e.g., CCFs, types of uncertainty) to more clearly navigate discussions of proxy assessments.

A complete outline of potential uncertainties and the often complex phenomena-measurement relationships is difficult to incorporate into grants, peer-reviewed manuscripts, and educational programs. The lack of extensive discussion of a proxy's uncertainty can lead to an over-simplification of these relationships (i.e., an under-consideration for CCFs and uncertainties). However, detailing how proxies might relate to some unknown CCFs (as is done

here) can make any proxy seem subject to countless unknown CCFs, which may engender an unwarranted dismissal of proxy data interpretations. Because proxy data informs models, manuscripts, and educational lessons, there needs to be a more universally accepted and functional means of discussing and conveying proxy uncertainty that is honest yet robust. Our spectrum of proxy measurements relates measurements to their CCFs, and thus the spectrum and

language provide such a means of conveying uncertainty in a universal way.

       Many studies, for example, have shown that $TEX_{86}$ trends were driven by changes in nitrogen availability and marine ecology in some paleo environments (Liu et al., 2009; Hurley et al., 2016; Junium et al., 2018, Polik et al., 2018). How can workers be sure that $TEX_{86}$ is not driven by these dynamics in other settings, unless those CCFs of nitrogen availability and marine

ecology changes are directly assessed? Because uncertainty in estimating these environmental characteristics are often not incorporated (as they are not incorporated in the current litany of quantitative $TEX_{86}$ calibrations; Table 1), we have described the *potential uncertainty* of $TEX_{86}$ (and other correlation-constrained proxies) as much higher than is often reported (Fig. 3b). By referring to $TEX_{86}$ as a *correlation-constrained proxy*, modelers, reviewers, and researchers can

immediately be aware of this under-reporting of uncertainty, which would inform their interpretation of the temperature estimates produced by $TEX_{86}$ in a meaningful yet succinct way.

## 3.3 Development of a proxy

Proxy development is the production and improvement of a calibration which quantitatively

accounts for all CCFs that contribute to the measured signal. The controlled characteristic of a mercury thermometer allows the measurement of temperature without needing an external calibration, as the temperature lines are calibrated to the exact expansion of mercury within the glass walls. Prior to the full calibration of the lines on the mercury thermometer, mercury might

have served as a proxy: a gram of mercury on a table would expand and contract with fluctuating

temperatures, which could be a qualitative, correlation-constrained proxy for temperature (the

mercury expanded, so the temperature likely got hotter).

Because proxy measurements do not account for the influence of all known CCFs,

quantitative proxy measurements require some external calibration equation to produce reliable

estimates. Calibrations express the relative effect of each causal factor (Fig. 2), and provides

insight into the applicability of a proxy by addressing the range in which the calibration is useful,

and the natural variability (uncertainty) associated with that calibration. Proxy applications are

limited to the range in which that proxy has been studied and calibrated; applications outside that

range do not produce reliable estimates.

Harold Urey's first description of the thermodynamic relationship between $\delta^{18}O_{calcite}$ and

calcite formation temperatures was simply "The calculated slope, 4.4 per mil between 0°C and

25°C" (Urey, 1948). More complex calibrations now exist for the $\delta^{18}O_{calcite}$ paleotemperature

proxy, which accounts for its numerous CCFs including $\delta^{18}O_{H2O}$ and biological effects (Ravelo

and Hillaire-Marcel, 2007; Hollis et al., 2019). While the $\delta^{18}O_{calcite}$ proxy is far from a controlled

measurement, its historical development exemplifies the consistent work to make proxies more

like a controlled measurements, i.e., to eliminate or limit the influence of CCFs. But what does

such proxy development look like in practice?

The first step of proxy development is the identification of some corollary relationship

between a measurable property (e.g., $\delta^{18}O$ of calcite) and property unable to be measured in a

controlled fashion (e.g., temperature of a past environment). At first order, these are usually

qualitative and based on some hypothesis to describe a system. Mercury expands with increasing

temperature due to general fluid dynamics; $^{18}O$ is more favorably incorporated into calcite at

lower temperatures due to differences in vibrational energies between $^{18}O$ and $^{16}O$; some

organisms alter their cell membranes to maintain homeostasis in variable environments.

Proxies that are based on such a corollary relationship can serve as *qualitative proxy*

*measures*, which provide useful comparative or relative information. This is the case for some

paleotemperature proxies: geological evidence of glacial expansion and retreat in a certain

location can indicate relative local temperature change, but variability in numerous (difficult or

impossible to constrain) CCFs prohibits a calibration to quantitative temperature changes in

degrees Celsius. Such comparative information is appropriate for many paleo studies, where the

question is focused on trends and relative changes through time or differences between sites. This corollary relationship can lead researchers into Harry Elderfield's "optimism phase," where the assumption of a direct, cause-effect relationship between a phenomena and an observation makes users optimistic that a proxy can be used with confidence (Elderfield, 2002).

        If researchers aim to use a proxy quantitatively, the relationship between the target

property (e.g., temperature), the observable property (e.g., $\delta^{18}O_{calcite}$), and all CCFs must be accounted for in a calibration (Fig. 2). Quantitative proxies require an (empirically derived) estimation or (logically deduced) inference of the influence of all CCFs represented in a calibration. Calcite precipitation experiments with variable pH, $\delta^{18}O_{H2O}$, salinity, and biomineralizing organisms have contributed to calibrations that factor in those CCFs, and

represent how they contribute to $^{18}O$ incorporation into calcite (Ravelo and Hillaire-Marcel, 2007). Studies using those calibrations must account for those CCFs. For example, calcite-producing organisms live in either bottom waters or surface waters – the temperature from the two will not only have slightly different CCFs, but will also reflect temperature from different parts of the water column. Workers would identify the type of organisms to know where it lived,

and would address the CCFs specific to that organism (e.g., Bemis et al., 1998). The process of testing CCFs must be extensive to provide confidence in the proxy. Often, this phase of development unearths unforeseen CCFs, such as the role of water column oxygenation on isoGDGT cyclicity (Qin et al., 2015; Hurley et al., 2016). While some have argued that this can lead to a "pessimism phase," where proxy users might no longer have confidence in that proxy's

utility (Elderfield, 2002), in fact these revelations are essential to proxy development – it is the scientific method at work, and such exhaustive testing of CCFs is a prerequisite for the confident use of a proxy.

        The identification and testing of CCFs is inherently an iterative processes. Urey and others provided serious consideration of CCFs before applying the $\delta^{18}O_{calcite}$ paleotempearture

proxy. It was proposed that the proxy be used only "if the isotopic composition of the water is known not to differ from the mean of the present seas, or…in the case that it does [differ], if both the isotopic composition of the carbonate and water are determined" (Urey et al., 1951). Urey described local variability in $\delta^{18}O_{H2O}$ due to evaporation and salinity as "the greatest difficulty" for accurate temperature measurements, but promised, "this problem is being studied from

several angles and it is hoped that corrections can be applied in the future" (Urey et al., 1951).

Urey's careful consideration of CCFs, and the subsequent and ongoing investigations into those CCFs, serves as an exemplar for proxy discussion, interpretation, and development.

Sometimes, the development of one proxy can constrain a CCF for another proxy by providing a new means of estimating that CCF. The development of the Mg/Ca paleotemperature proxy, based on the incorporation of magnesium relative to calcium in foraminiferal calcite, provided an independent constraint on temperature at the same time (i.e., mid-1990s) that $\delta^{18}O_{calcite}$ was being developed as a paleotemperature proxy (Hastings et al., 1998). By using Mg/Ca to estimate temperature in the same setting as $\delta^{18}O_{calcite}$, researchers were able to independently constrain temperature, and thus use $\delta^{18}O_{calcite}$ to estimate $\delta^{18}O_{H2O}$ (Mashiotta et al., 1999). The development of two independent paleothermometers, each with their own CCFs, provided researchers new opportunities and greater confidence in applying those proxies; $\delta^{18}O_{calcite}$ and Mg/Ca combined helped to identify the degree to which $\delta^{18}O_{H2O}$ influenced the $\delta^{18}O_{calcite}$ proxy, and resulted in a new means to constrain the CCF of $\delta^{18}O_{H2O}$ for future studies. Similarly, multiple studies have compared temperature estimates from $TEX_{86}$ as well as other organic (e.g., alkenones; Huguet et al., 2006; Lee et all., 2008; Li et al., 2013) and inorganic (e.g., Mg/Ca and $\delta^{18}O_{calcite}$; e.g., Hollis et al., 2012; Hetzberg et al., 2016; O'Brien et al., 2017) proxies in the same settings. While those multi-proxy comparative studies are helping to identify CCFs related to $TEX_{86}$ and other paleotemperature proxies, the numerous unconstrained CCFs related to $TEX_{86}$ make direct testing of CCFs difficult for even those comparative studies. For example, are deviations between $\delta^{18}O_{calcite}$ and $TEX_{86}$ due to depth of production in the water column (e.g., Li et al., 2013; Hetzberg et al, 2016), production season (Huguet et al., 2006), or some other CCF like nutrient availability (Hurley et al., 2016)? Some $TEX_{86}$ applications have used independent proxies to constrain CCFs related to the environment, such as the use of the BIT index (Hopmans et al., 2004) to estimate changes in the input of isoGDGTs from non-marine sources (e.g., Weijers et al., 2006; Hollis et al., 2012). Future work integrating the *physiological* CCFs associated with $TEX_{86,}$ such as changes in water column oxygenation (Qin et al., 2015) and nutrient availability (Hurley et al., 2016) into such multi-proxy comparisons would better constrain the role of different CCFs on $TEX_{86}$ paleotemperature estimates.

Alternatively, the use of statistical methods can elucidate CCFs and their impact on proxy measurements. One example is the Bayesian statistical modelling approach, which uses existing data (usually field-produced calibrations) over a wide range of environments to produce a "best-

fit" calibration for the range of values measured in a given study. The resulting model allows workers to identify which environments/locations produce a calibration that best fits their data, and thus provides a means for workers to investigate environmental conditions, and the related

CCFs, that more fully express the relationship between, for example, $TEX_{86}$ and temperature (Tierney and Tingley, 2014). In fact, the PMIP proxy assessment (Hollis et al., 2019) recommends $TEX_{86}$ users utilize the Bayesian calibration fit as the best current means to estimate paleotemperatures (Hollis et al., 2019), demonstrating how the field may use these statistical methods to provide best-practices for measurement applications. Similarly, stochastic modelling

approaches are used in hydrological data interpretations as a means to estimate the partial effects (or confounding effects) of different causal factors contributing to a given signal (Yevjevich, 1987), and such approaches could be utilized by the paleotemperature community.

            Alternatively, the application of transfer functions, including Proxy System Models, are used to make inferences about CCFs. Transfer functions provide a theoretical (rather than

empirical) constraint on a systems' properties in an attempt to predict the quality of properties rather than observe them (Telford and Birks, 2005). While the reliability of transfer functions itself is an area of active discussion (e.g., Telford et al., 2004; Telford et al., 2013), transfer functions represent yet another statistical approach used to account for CCFs in lieu of empirical observations, and are employed by some to reduce uncertainty for inference-constrained proxies.

For example, Proxy System Models use transfer functions to provide an assessment of proxy-phenomenon relationships, and the driving mechanisms behind proxy measurement outputs (e.g., Dee et al., 2016; Dee et al., 2018; Okazaki and Yoshimura, 2019). These statistical methods are an important aid in the determination of CCFs on observational signals, and can be powerful in the development of proxy calibrations.

Ultimately, a mix of variable-controlled laboratory experiments, statistical analyses, and field validation experiments all contribute to proxy development. The identification and expression of corollary relationships in a statistical regression is only the first step. Comparisons between laboratory (e.g., culture) experiments and field measurements might produce different calibrations; causes for differences in the regression should be investigated. For $TEX_{86}$, the

recognition of significant variability amongst field calibrations led workers to investigate non-temperature properties, such as physiological effects of Thaumarchaeota, in variable-controlled in-laboratory culture experiments (e.g., Elling et al., 2014; Qin et al., 2015; Hurley et al., 2016).

In response, field studies of isoGDGT cyclization were performed in modern and paleo settings (e.g., Hurley et al., 2016; Junium et al., 2018; Polik et al., 2018), and compared with those CCFs identified in culture experiments. These studies together suggest that $TEX_{86}$ users should aim to measure changes in water column oxygenation, ammonia availability, and ecosystem structure, and incorporate those measurements quantitatively into a calibration to develop $TEX_{86}$ as an observation-constrained proxy. Unfortunately, the current limitation (and area of most research) concerns the production of a calibration which accurately reflects all CCFs (Table 1). Many researchers have moved forward with applying $TEX_{86}$ in paleo studies, providing an in-text inference of some CCFs, often concluding that the CCFs do not affect the temperature estimate (e.g., O'Brien et al., 2017), or independently measuring a select number of CCFs (such as changes in the input of isoGDGts using the BIT index; e.g., Weijers et al., 2006). The lack of a unifying calibration that quantitatively accounts for those CCFs implies that these applications exemplify *correlation-constrained proxy* measurements, and the associated reported uncertainty should aim to reflect the accompanying potential uncertainties (Fig. 3b).

Because an ideal calibration reflects all contributing pieces of a system (Fig. 2), *a single calibration is necessary for a proxy to be reliably quantitative.* It should be verifiable and applicable in a wide variety of locations, times, and situations. If the calibration is inadequate for some situation, then the calibration does not account for *all* potential CCFs. We consider these calibrations incomplete; for some systems, the unknown CCF does not change, and the calibration explains the corollary relationship, but for other systems, the unknown CCF is introduced or changes, such that the calibration no longer adequately represents the relationship between the measured entity and the property in question. This is the state of current $TEX_{86}$– each different calibration purports a different quantitative description of the relationship between causal factors (e.g, temperature) and isoGDGT cyclicity (Table 1), and none quantitatively account for CCFs (Table 1; Fig. 3a). Ongoing work to better constrain what CCFs are at play, and how they can be quantified, can move $TEX_{86}$ towards a more observation- or inference-constrained proxy, and lead to more reliable $TEX_{86}$ paleotemperature estimates.

While we use $TEX_{86}$ as an exemplar here, we recognize that limitations in quantitative proxy development and calibration exist across all fields of study, and particularly in the Earth sciences. Not all proxies need be quantitative, and all quantitative proxies present uncertainty. But for a measurement to be most effective (broad applications, less uncertainty), it should be

developed as close to a controlled measurement as possible. This means developing a causal,

mechanistic understanding of the relevant system (i.e., a single calibration) as a means to

adequately control for the influence of CCFs and produce reliable proxy estimates.

## 4 Conclusions

The distinction between controlled and proxy measurements, and within proxy measurements,

serves a more functional role for interpreting, assessing, and developing proxies than previous

distinctions between proxy and "direct" measurements. The language proposed here concerning

proxy calibrations (e.g., observation- versus inference-constrained proxy) and uncertainty (e.g.,

analytical versus potential) succinctly and directly addresses the relationship between

measurements and the property they intend to describe, and more clearly directs proxy

calibration development.  Using this language, modelers can more confidently appropriate proxy

data outputs into their models, researchers can more efficiently design studies to produce robust

measurements, reviewers can more easily assess the reporting of uncertainty and interpretations,

and educators can more clearly convey the differences in measurements available for students to

learn from, apply, and improve. Readers may find that observational measurements not typically

considered proxy measurements in their field may in fact fall on the proxy end of our spectrum.

We hope that such realizations might drive workers to investigate what has been taken for

granted in previous interpretations, or how future study designs can more accurately assess   and

account for CCFs. Ultimately, we propose that as much can be learned about a system by

developing a proxy as can be learned by applying it.



# Glossary of terms

- **Confounding Causal Factors (CCFs):** Characteristics of an environment that affect the output of a measurement, but are not the property being measured.

- **Controlled measurement:** Measurement that has been manufactured/designed to eliminate the potential effects of all known CCFs on the measurement output.

- **Proxy measurement:** Measurement that does not eliminate the influence of all known CCFs on the intended/targeted property.

- **Observation-constrained proxy:** Proxy measurement where the CCFs are quantitatively incorporated into a calibration, and are accounted for with values produced by other proxy measurement estimates or controlled measurements.

- **Inference-constrained proxy:** Proxy measurement where the CCFs are quantitatively incorporated into a calibration, and are qualitatively accounted for using a reasoned approximation (inference) of the value based on comparisons to similar systems, rather than values produced by measurements of the system in question.

- **Correlation-constrained proxy:** Proxy measurement that does not account for known
CCFs, but is based on a hypothesized relationship between a certain property and a measurement output. Uses a calibration that does not quantitatively represent the causal structure of the system.

- **Analytical uncertainty:** The uncertainty associated with the precision and accuracy of the analytical instrument.

- **Potential uncertainty:** The degree to which the measurement/estimated value is affected by something other than the property being measured.

- **Reported uncertainty:** A textual and/or numerical representation of the combined analytical and potential uncertainties associated with a measurement.


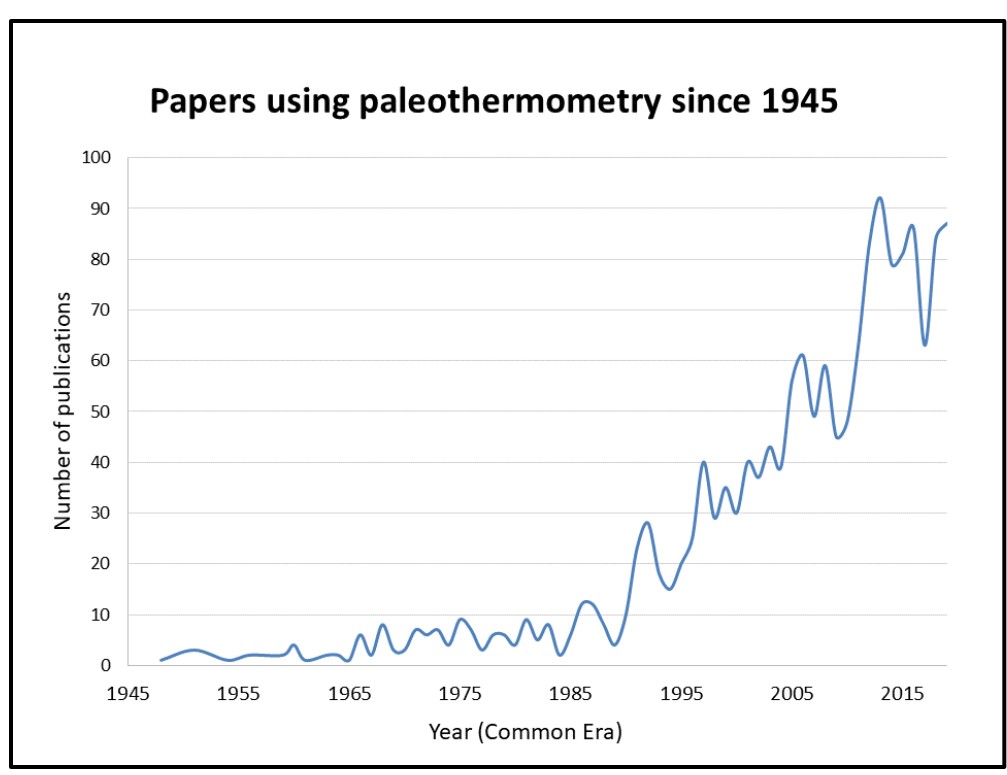

Figure 1: Papers discussing paleotemperature proxies since 1945, from a Web of Science database query of "articles" and "reviews" for topics "Paleothermometry OR Paleothermometer OR Paleotemperatures."


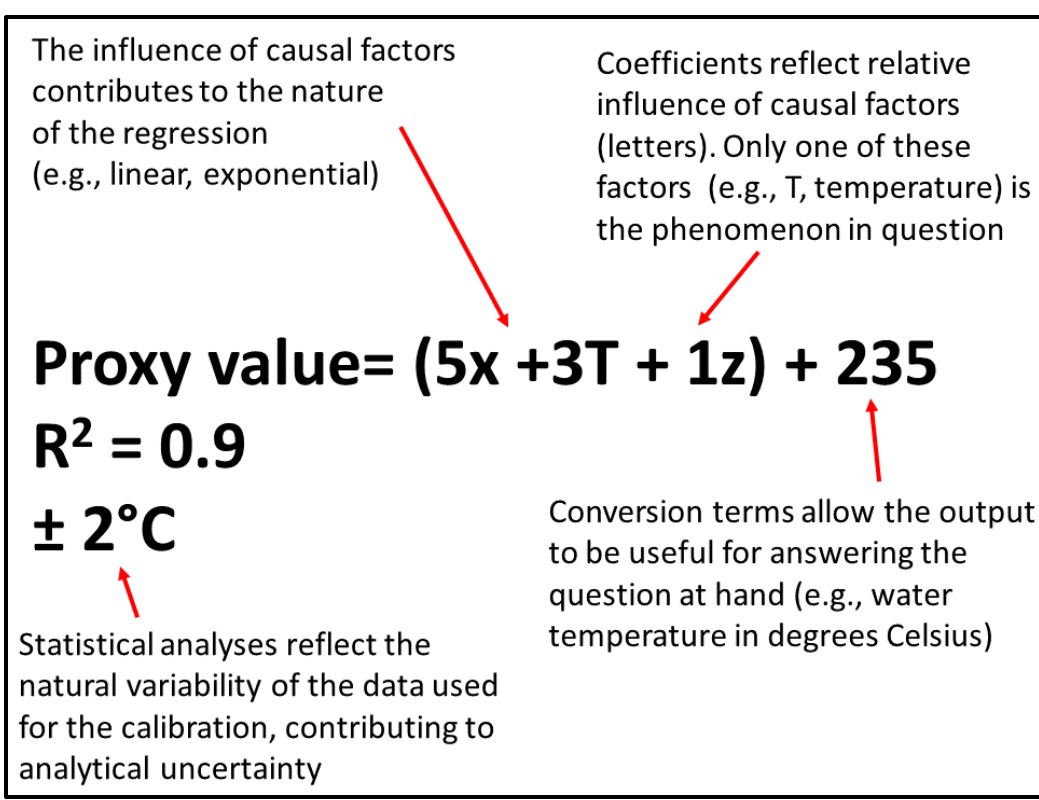

Figure 2: Schematic and description of an idealized calibration for a hypothetical paleotemperature proxy.


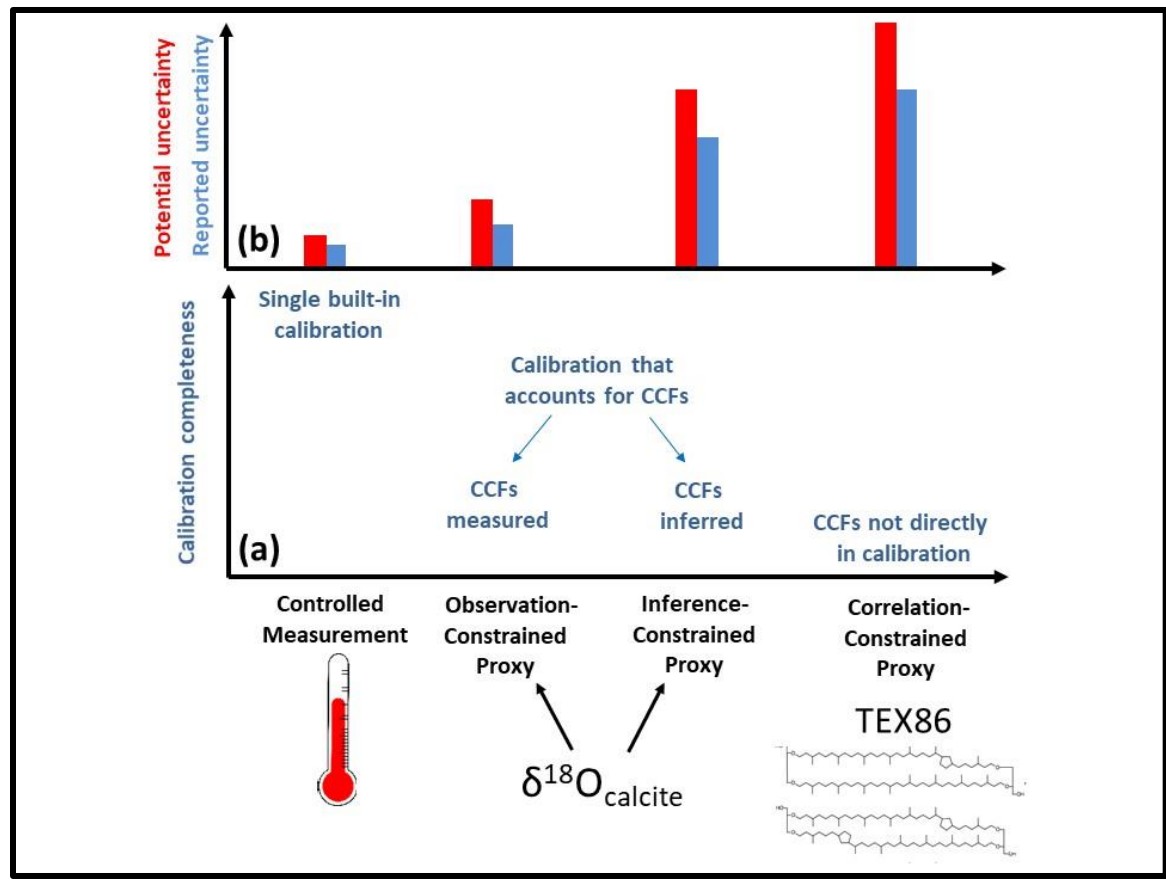

Figure 3: A spectrum (X axis) of observational measurements as function of their incorporation of confounding causal factors and related uncertainty. (a) Bottom Y axis describes the completeness of a measurement's calibrations (i.e., how completely a calibration accounts for all causal factors). Controlled measurements on the left have full control of all causal factors.

Observation-constrained proxies have a calibration that quantitatively accounts for CCFs, and allows the researcher to measure those CCFs. Inference-constrained proxies also have a calibration that quantitatively accounts for CCFs, but the researcher cannot measure the CCFs, so the quantitative values for CCFs used in the calibration must be inferred from other evidence. On the right, correlation-constrained proxies have the least direct (quantitative) control of the causal

factors, with calibrations that do not quantitatively account for CCFs.  (b) Top Y axis represents uncertainty of each measurement, with the red line signifying potential uncertainty and the blue bar showing range of reported uncertainty in literature. Because analytical uncertainty varies greatly between proxies, instruments, and users, we have excluded its representation. The wide range of reported uncertainty (blue bars) derives from the wide range of reported uncertainty

associated with each measurement in existing literature.

| Range (°C) | Equation | Reference |
| --- | --- | --- |
| 0-30 | $T = (TEX_{86} - 0.27) / 0.015$ | Schouten et al. (2002) |
| 22-30 | $T = (TEX_{86} - 0.016) / 0.027$ | Schouten et al.(2003) |
| 10-28 | $T = (TEX_{86}' - 0.2) / 0.016$ | Slujis et al. (2006) |
| 5-30 | $T = -10.78 + 56.2 \times TEX_{86}$ | Kim et al. (2008) |
| 25-28 | $T = (TEX_{86} + 0.09) / 0.035$ | Trommer et al. (2009) |
| -3-30 | $T = 50.475 - 16.332 \times (1/TEX_{86})$ | Liu et al. (2009) |
| -3-30 | $T = 81.5 \times TEX_{86} - 26.6$ | Kim et al. (2010) |
| -3-30 | $T = -19.1 \times (1/TEX_{86}) + 54.5$ | Kim et al. (2010) |
| -3-30 | $T = 49.9 + 67.5 \times (GDGT\ index\text{-}1)$ | Kim et al. (2010) |
| 5-30 | $T = 38.6 + 68.4 \times (GDGT\ index\text{-}2)$ | Kim et al. (2010) |
| 10-40 | $T = 48.2 \times TEX_{86} + 1.04$ | Kim et al. (2010) |
| 10-40 | $T = -9 \times (1/TEX_{86}) + 45.2$ | Kim et al. (2010) |
| 10-40 | $T = 42.9 \times (GDGT\ index\text{-}1) + 46.5$ | Kim et al. (2010) |
| 10-40 | $T = 52 \times (GDGT\ index\text{-}2) + 42$ | Kim et al. (2010) |
| 4-30 | $T = -14 + 55.2 \times TEX_{86}$ | Powers et al. (2010) |
| 10-30 | $T = 3.5 + 38.9 \times TEX_{86}$ | Tierney et al (2010) |
| -2-30 | $T = (TEX_{86} - 0.3038) / 0.0125$ | Shevenell et al. (2011) |
| 14-34 | $T = 32.873 \times \ln(GDGT\ index\text{-}1) + 50.771$ | Hollis et al. (2012) |
| 14-34 | $T = 39.036 \times \ln(TEX_{86}) + 36.455$ | Hollis et al. (2012) |
| 15-35 | $T = (TEX_{86} - 0.21) / 0.015$ | Qin et al. (2015) |
| 10-30 | $TEX_{86} = -0.0006T^2 + 0.023T + 0.33$ | Qin et al. (2015) |
| 10-25 | $TEX_{86} = -0.0017T^2 + 0.054T + 0.11$ | Qin et al. (2015) |
| 2-10 | $T = 27.898(TEX_{86}^L) + 22.723$ | Harning et al. (2019) |

| Name | Calculations | Reference |
| --- | --- | --- |
| $TEX_{86}$ | [GDGT-2]+[GDGT-3]+[Cren']/[GDGT-1]+[GDGT-2]+[GDGT-3]+[Cren'] | Schouten et al. (2002) |
| $TEX_{86}'$ | [GDGT-2]+[GDGT-3]+[Cren']/[GDGT-1]+[GDGT-2]+[Cren'] | Slujis et al. (2006) |
| $TEX_{86}^L$ | $-\log$([GDGT-2]/[GDGT-1]+[GDGT-2]+[GDGT-3]) | Kim et al. (2010) |
| $TEX_{86}^H$ | $0.99 \times TEX_{86}^L + 0.12$ | Kim et al. (2010) |
| GDGT index-1 | $\log$([GDGT-2]/[GDGT-1]+[GDGT-2]+[GDGT-3]) | Kim et al. (2010) |
| GDGT index-2 | $\log(TEX_{86})$ | Kim et al. (2010) |


Table 1: Compilation of $TEX_{86}$ calculations and calibrations as of 2020. Modified from Tierney (2012).

## Author contribution

JW and FGB designed the study. FGB wrote the manuscript, and JW and FGB edited the manuscript.

## Data Availability

All data described here is presented in previously published literature and is cited as such.


## Competing interests

The authors declare that they have no conflict of interest.

## Acknowledgements

We thank the Center for the Study of Origins for their support. We thank T. Marchitto, G. Miller, B. Johnson, M. Huber, and F.D. Boudinot for their helpful comments that improved the manuscript, and H. Spero, J. Sepúlveda, and C. Cleland for their discussions that improved the manuscript. FGB acknowledges the Department of Geological Sciences at the University of Colorado Boulder, NSF Division of Earth Sciences Earth-Life Transitions (ELT) program grant

#1338318, and the American Chemical Society Petroleum Research Fund (ACS-PRF) - Doctoral New Investigator Award #58815-DNI2 for their support. JW acknowledges the Department of Philosophy and the Graduate School at the University of Colorado Boulder for their support.

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
