# Peer review of "Does a proxy measure up?: A framework to assess and convey proxy reliability"

_Climate of the Past, 2020_

## Referee Comment (RC1) · Anonymous Referee #1 · 7 Apr 2020

General Comments

The manuscript provides an epistemological discussion about paleotemperature proxies used by earth system scientists. The paper suggests that for proxies geared toward reconstructing past temperature, there should be a more coherent and consistent way to acknowledge the confounding causal factors (CCFs) that potentially complicate all available proxies, such as $\delta$18O calcite, TEX86, and mercury thermometers.

The points are good reminders for paleoclimatologists, and the paper is technically sound and clearly written. Still, I don't find that the manuscript offers any truly novel ideas or any tangible way of tackling the uncertainties in proxy systems. Nor do I see a

clear and unified suggestion for how proxy uncertainties should be discussed in future papers. Most paleoclimatologists are well aware of the shortcomings of the proxies, and researchers are working to incorporate CCFs into Proxy System Models. The paper might be difficult for non-experts to find useful.

Specific Comments

I consider there to be a distinction between "proxy" and "indicator", where proxies offer a quantitative estimate of past environmental condition, whereas indicators give a non-quantitative description of past changes. I think this dichotomy also deserves a place in this discussion. At line 478, I think that by definition, proxies are quantitative because they are substituting for a variable of interest.

Given that the paper discusses the importance of word choice in describing proxies, I'm surprised they used the word "paleothermometer" so freely. In my view, this word and its relatives are sometimes used inappropriately. Its use in this manuscript detracts from the discussion on the understanding of the proxy interpretations. I recommend using "paleotemperature proxy" rather than "paleothermometer" for the very reason they discuss – that there are confounding factors and the proxies are proxies, not old thermometers.

It seems like this paper revisits the utility of transfer functions which has received considerable discussion, in, for example, Journal of Paleolimnology. Consider the numerous transfer functions derived for diatom assemblages – can a single diatom record really be used to estimate salinity, DOC, temperature, phosphorus, etc., or, does this example approach the case where many CCFS are addressed for a single proxy system? I was surprised that the term "transfer function" was never used in this manuscript, given that this is the standard method by how we translate any proxy measurement into paleotemperature estimates.

Along those lines, there are numerous papers evaluating how different assumptions, different statistical methods, independence of datasets, autocorrelation, etc. all impact

reported uncertainties in transfer functions. I suggest the authors incorporate discussion or references to these earlier works. For example:

Telford, R. J., & Birks, H. J. B. (2005). The secret assumption of transfer functions: problems with spatial autocorrelation in evaluating model performance. Quaternary Science Reviews, 24(20-21), 2173-2179.

Telford, R. J., Andersson, C., Birks, H. J. B., & Juggins, S. (2004). Biases in the estimation of transfer function prediction errors. Paleoceanography, 19(4).

Telford, R. J., Li, C., & Kucera, M. (2013). Mismatch between the depth habitat of planktonic foraminifera and the calibration depth of SST transfer functions may bias reconstructions. Climate of the Past, 9(2), 859.

Guiot, J., & De Vernal, A. (2011). Is spatial autocorrelation introducing biases in the apparent accuracy of paleoclimatic reconstructions?. Quaternary Science Reviews, 30(15-16), 1965-1972.

Of particular note is the recent work done by Bronwen Konecky, Sylvia Dee, and many others to develop Proxy System Models (PSM). With a push to incorporate proxy measurements into climate models, it seems timely to discuss how understanding and modeling the CCFs is particularly important for PSM development.

Why is there so much focus given to mercury thermometers, but none given to digital thermometers? While mercury thermometers are described as proxy-like, do the authors suggest there is some way of measuring temperature that is not a proxy? What is the true end member there or does it not exist?

Line 190: I would like to see another example of an inference-constrained proxy to make sure that concept is clear, because it is still not obvious to me after this one example how it differs from the observation-constrained proxy.

Line 342: Measurements do not require external calibration equations. The translation of measurements into temperature estimates do.

Line 409: "This allowed users to apply d18Ocalcite..." needs to clarify that the use is no longer to estimate past temperatures but to estimate past seawater d18O. They should specify what opportunities and confidence grew from the combining of Mg/Ca and d18O?

In Figure 3b, since the data come from reported uncertainties in the literature, it seems like it would be possible to include values on the y-axis. I would like to see those values and how they compare between these systems.

---

## Referee Comment (RC2) · Julie Griffin (Referee) · 27 Apr 2020

The manuscript by Boudinot and Wilson represents a substantial contribution to paleoclimate research by describing a new system for presenting proxy data that explicitly identifies potential inaccuracies. By addressing how scientist reconstruct ancient climates, the manuscript is appropriate for "Climates of the Past." The topic of proxy reliability is something all paleoclimatologists are aware of and wrestle with, but not something for which we have a universal language. A universal framework for reporting proxy reliability is necessary as we try to synthesize the large mass of data collected into coherent datasets for making global conclusions. The differences in the

same types of paleoclimate data and their uncertainties are reported forces scientists into comparing apples and oranges. The text lays out an example of this in the paragraph beginning on line 272. Without the ability to compare similar datasets, such as temperatures produced from TEX-86 analysis, we are limited in the breadth of our conclusions. More critically, this issue prevents us from independently assessing the reproducibility of conclusions.

The manuscript is difficult to review for Scientific Quality since it is not a traditional analysis of original data nor a review paper. The text does contain a logical breakdown of what makes proxies unreliable: variation based on context. The manuscript also presents many new terms (confounding causal factors or CCFs, observation-constrained proxy, inference-constrained proxy, correlation-constrained proxy) that need explicit definitions to be clear. I suggest use of a glossary. The evaluation framework presented does not assess the magnitude of influence of different CCFs on the proxy interpretations. For example, air temperature may influence the seawater temperature recorded by foram oxygen isotopic composition, but not beyond the analytical error of the oxygen isotopic measurement. If this is the case, then does the lack of calibration for air temperature mean that the seawater temperature is more poorly constrained? The text is also missing a detailed consideration of how geologic age influences proxy reliability. The issue is mentioned (ln 138-142), but not explored as a significant CCF. Reliability with geologic age may not be possible to integrate into a calibration. My reading of the manuscript indicates that the authors assume that CCFs may always be known, which isn't possible. Determining if this proxy-assessment framework is useful can really only be determined by applying the framework to other proxies, which is beyond the scope of this paper.

The presentation of the manuscript is clear for an abstract subject. The title reflects the contents of the paper, the language is fluent, the document is well structured, and the conclusions are well referenced. The abstract is missing an explanation of CCFs, which are the critical subject of the rest of the paper. Figure three does need some

clarification. The y-axis of figure 3b needs to be defined - low uncertainty to high uncertainty or absolute values of uncertainty. Also, the sizes of the boxes seem to contradict the text as ln 270 states that potential uncertainty should always be higher than reported uncertainty, but then the blue "reported uncertainty" box is larger than the potential uncertainty box. Figure 3b is critical for understanding the framework but currently very unclear.

I've included a PDF of the manuscript with my comments for further investigation.

Please also note the supplement to this comment:
https://www.clim-past-discuss.net/cp-2020-12/cp-2020-12-RC2-supplement.pdf

**Supplement:**

[revised manuscript text omitted]

---

## Author Comment (AC1) · 10 Jun 2020

We thank this anonymous referee for their thoughtful comments, which highlight the applicability of our manuscript to the Climate of the Past audience and provide suggestions to improve the scope of the paper.

We use a suite of temperature measurements and proxies as case studies to demonstrate that uncertainties associated with proxies could be better represented, and potentially accounted for. We provide specific language to guide such representation, including the novel terms confounding causal factor, controlled measurement, and inference- and observation-constrained proxy. While we agree that most paleoclimatol-

ogists are well aware of these ideas, the lack of codified language to represent these ideas has remained a barrier to open discussion of the subject. As such, we do provide clear suggestions for future discussions of proxy uncertainty. While our goal is not to provide a "tangible way of tacking the uncertainties in proxy systems," we do see that as an objective that could be met using our proposed framework in the future.

The referee provides many opportunities to improve/clarify our discussion and expand the scope of our work. For example, we agree that paleotemperature proxy is a useful replacement of the term paleothermometer. While our explicit in-text definition of paleothermometer (lines 65-66) situates our use of the term throughout the text, paleotemperature proxy can easily be substituted, and indeed highlights the proxy association that is crucial to our discussion. We will modify the revised manuscript accordingly.

We are also happy to include the provided references on transfer functions and proxy system models, which indeed have contributed to the assessment of uncertainties and CCFs during proxy development. The incorporation of those will certainly help to demonstrate the rich history of existing methods for developing proxies and reducing the effect of unknown CCFs.

We will certainly include further discussion and examples of inference-constrained proxies as they relate to observation-constrained proxies in the revised manuscript. Similarly, a "glossary" of proposed terms as suggested by another referee should help communicate the distinction between the two.

Other points of clarification needed in the text, including lines 342 and 409, are well-received and will be modified accordingly in the revised manuscript.

While we feel that an extensive discussion of digital thermometers is not necessary for our case-study approach to describing our framework, we are happy to briefly incorporate digital thermometers by situating them on our spectrum where appropriate. Indeed, it fits within our framework and helps to provide context to the proposed framework.

We are happy to revise Fig. 3b to improve its clarity. However, as noted in the text (lines 272-290), the existing literature currently lacks a unified assessment of uncertainty, limiting the construction of a quantitative y-axis. Modifications to the figure for the revised manuscript will, however, include y-axis descriptors to clarify the ideas presented there.

Finally, while the reviewer may consider a distinction between proxy and indicator, we see the latter as too broad a term. Our distinction between quantitative and qualitative proxies serves the same function, and is more clear than introducing a new, broad term such as indicator. We do appreciate this comment, however, as it helps to focus and clarify our language used throughout the text.

―――――――――――――――

---

## Author Comment (AC2) · 10 Jun 2020

We thank Dr. Griffin for the comments on our manuscript, which describe the utility and novelty of our work for the Climate of the Past readership, while also offering suggestions to improve the clarity and focus of our discussion.

We love the idea of a glossary to explicitly define the terms proposed in our text, which would certainly aid in the dissemination of our ideas. We are happy to include that in the revised manuscript. Additionally, we will add an explicit explanation of CCFs in the abstract to highlight this major theme of our work.

The issue of geologic time as it influences CCFs is indeed discussed in our manuscript (lines 236), though we agree that such an important aspect of our discussion needs to be mentioned earlier in the discussion, and with more explanation. We will make those changes in the revised manuscript. Similarly, while we do discuss the existence of unknown CCFs (line 266), this idea will be mentioned earlier in our discussion in the revised manuscript, and will also be incorporated into the glossary definition of CCFs. Importantly, it is the ubiquity of unknown CCFs in all proxies that makes potential uncertainty always greater than that reported, highlighted in Fig. 3b.

We appreciate the suggestions to improve Fig. 3b. The revised manuscript will include added descriptors for the y-axis, and new representations (rather than the blue and red bars) of the magnitude of those uncertainties. This should help clarify the relationship between potential and reported uncertainties shown in Fig. 3b.

We agree with the reviewer that our manuscript "does not assess the magnitude of influence of different CCFs on the proxy interpretations;" we see that as specific to individual proxy systems, and the subject of more individual proxy-specific work than our manuscript. For example, Hollis et al. (2019) do indeed describe the magnitude of different CCFs for some proxies – though they lack the term CCF to guide their discussion. We hope that our manuscript provides a framework to encourage and improve such assessments for all forms of measurement in future work.

The PDF comments are very helpful in gauging which sections require rewording or further discussion, and which sections currently read strong and serve as important aspects to convey our ideas. Areas in need of clarifying text will be modified accordingly in the revised manuscript. We recognize and appreciate the thoughtfulness put into those comments.
* * *

---

## Author Response (AR1)

We are pleased to submit a revised version of our manuscript entitled "Does a proxy measure up?: A framework to assess and convey proxy reliability" for further consideration at Climate of the Past. Comments from two reviewers and the editor were very positive, and suggested several areas for improvement by adding additional discussion and clarifying some important text. The additions and changes in the present manuscript take into account those recent constructive comments, providing a more robust discussion of our work. Specific suggestions are described below in black text, with our description of how they were accounted for outlined below in red text.

Replace paleothermometer with paleotemperature proxy.

This has been modified accordingly throughout the text.

Add text and references on transfer functions and proxy system models

We have incorporated these into the discussion on existing methods to assess properties and uncertainties associated with CCFs during proxy development and applications. Specifically, we have added a paragraph (lines 483-498) describing how these are used, and have added the following references. We appreciate this suggestion, which allows us to more thoroughly discuss existing efforts to assess and account for CCFs.

Dee, S.G., Russell, J.M., Morrill, C., Chen, Z., and Neary, A.: PRYSM V2.0: A proxy model for lacustrine archives, Paleoceanography and Paleoclimatology, 33, 1250-1269, https://doi.org/10.1029/2018PA003413, 2018.

Dee, S.G., Steiger, N.J., Emile-Geay, J., Hakim, G.J.: On the utility of proxy system models for estimating climate states over the common era, Journal of Advances in Modeling Earth Systems, 8, 1164-1179, https://doi.org/10.1002/2016MS000677, 2016.

Okazaki, A. and Yoshimura, K.: Global evaluation of proxy system models for stable water isotopes with realistic atmospheric forcing, JGR Atmospheres, 124, 8972-8993, https://doi.org/10.1029/2018JD029463, 2019.

Telford, R.J., and Birks, H.J.B.: The secret assumption of transfer functions: problems with spatial autocorrelation in evaluating model performance, Quaternary Science Reviews, 24, 2173-2179, 10.1016/j.quascirev.2005.05.001, 2005.

Telford, R.J., Andersson, R., Birks, H.J.B., and Juggins, S.: Biases in the estimation of transfer function prediction errors, Paleoceanography, 19, PA4014, https://doi.org/10.1029/2004PA001072, 2004.

Telford, R.J., Li, C., Kucera, M.: Mismatch between the depth habitat of planktonic foraminifera and the calibration depth of SST transfer functions may bias reconstructions, Climate of the Past, 9, 859-870, https://doi.org/10.5194/cp-9-859-2013, 2013.

Add further discussion of inference-constrained proxies as they relate to observation-constrained proxies.

We have added extended text (three new paragraphs) describing the differences between observation- and inference-constrained proxies (lines 209-238). This new discussion details how the two differ in theory and practice, and how that manifests as difference reliability and uncertainty, making more clear how the two differ.

A "glossary" of proposed terms as suggested by another referee should help communicate the distinction between the two.

We have added a glossary at the end of the text (lines 570-594), and have included reference to it throughout when new terms are introduced. We appreciate this suggestion, as we agree that it makes the new terms more digestible and memorable.

Line 342: Referee stated "Measurements do not require external calibration equations. The translation of measurements into temperature estimates do"

We have added to this sentence (lines 387-389) to clarify this point.

Line 409: clarification needed to differentiate between d18O of seawater and calcite, and relationship between Mg/Ca and d18Ocalcite

We have added text (lines 455-458) to make this relationship more clear, defining the exact advances made by the combination of Mg/Ca and d18Ocalcite.

Briefly incorporate digital thermometers by situating them on our spectrum where appropriate

We have added a new paragraph (lines 115-123) to incorporate digital thermometers as another point of reference for controlled measurements. We appreciate this suggestion, as it will further reinforce the use and scope of our proposed spectrum of measurements, and allows us to more fully develop our case study within temperature measurements.

Revise Fig. 3b to improve its clarity.

We have modified this panel to more clearly show the relationship between reported and potential uncertainty by plotting the two as bars.

Give some additional thought to the term "indicator"

We appreciate this suggestion, as the lack of consistency in the use of this term in the literature highlights the motivations of our study. We have included a new paragraph (lines 138-152) that illustrates how this term has been used, and describes how it fits within our spectrum and further justifies our study.

Issue of geologic time as it influences CCFs

We have added text (lines 166-172) to make more clear how the influence of geologic time influenced researcher's ability to constrain CCFs during proxy applications, and how that influences overall uncertainty associated with proxy measurements.

Add explicit mention of CCFs in abstract

We have made this change accordingly (lines 28-29).

[revised manuscript text omitted]

The influence of causal factors contributes to the nature of the regression (e.g., linear, exponential)

Coefficients reflect relative influence of causal factors (letters). Only one of these factors (e.g., T, temperature) is the phenomenon in question

**Proxy value= (5x +3T + 1z) + 235**
**$R^2$ = 0.9**
**± 2°C**

Conversion terms allow the output to be useful for answering the question at hand (e.g., water temperature in degrees Celsius)

Statistical analyses reflect the natural variability of the data used for the calibration, contributing to analytical uncertainty

Figure 2: Schematic and description of an idealized calibration for a hypothetical paleotemperaturethermometer proxy.

[Figure]

[Figure]

Figure 3: A spectrum (X axis) of observational measurements as function of their incorporation of confounding causal factors and related uncertainty. (a) Bottom Y axis describes the completeness of a measurement's calibrations (i.e., how completely a calibration accounts for all causal factors). Controlled measurements on the left have full control of all causal factors. Observation-constrained proxies have a calibration that quantitatively accounts for CCFs, and allows the researcher to measure those CCFs. Inference-constrained proxies also have a calibration that quantitatively accounts for CCFs, but the researcher cannot measure the CCFs, so the quantitative values for CCFs used in the calibration must be inferred from other evidence. On the right, correlation-constrained proxies have the least direct (quantitative) control of the causal factors, with calibrations that do not quantitatively account for CCFs. (b) Top Y axis represents uncertainty of each measurement, with the red line signifying potential uncertainty and the blue bar showing range of reported uncertainty in literature. Because analytical uncertainty varies greatly between proxies, instruments, and users, we have excluded its representation. The wide range of reported uncertainty (blue bars) derives from the wide range of reported uncertainty associated with each measurement in existing literature.

| Range (°C) | Equation | Reference |
|---|---|---|
| 0-30 | $T = (TEX_{86} - 0.27) / 0.015$ | Schouten et al. (2002) |
| 22-30 | $T = (TEX_{86} - 0.016) / 0.027$ | Schouten et al.(2003) |
| 10-28 | $T = (TEX_{86}' - 0.2) / 0.016$ | Slujis et al. (2006) |
| 5-30 | $T = -10.78 + 56.2 \times TEX_{86}$ | Kim et al. (2008) |
| 25-28 | $T = (TEX_{86} + 0.09) / 0.035$ | Trommer et al. (2009) |
| -3-30 | $T = 50.475 - 16.332 \times (1/TEX_{86})$ | Liu et al. (2009) |
| -3-30 | $T = 81.5 \times TEX_{86} - 26.6$ | Kim et al. (2010) |
| -3-30 | $T = -19.1 \times (1/TEX_{86}) + 54.5$ | Kim et al. (2010) |
| -3-30 | $T = 49.9 + 67.5 \times (GDGT\ index\text{-}1)$ | Kim et al. (2010) |
| 5-30 | $T = 38.6 + 68.4 \times (GDGT\ index\text{-}2)$ | Kim et al. (2010) |
| 10-40 | $T = 48.2 \times TEX_{86} + 1.04$ | Kim et al. (2010) |
| 10-40 | $T = -9 \times (1/TEX_{86}) + 45.2$ | Kim et al. (2010) |
| 10-40 | $T = 42.9 \times (GDGT\ index\text{-}1) + 46.5$ | Kim et al. (2010) |
| 10-40 | $T = 52 \times (GDGT\ index\text{-}2) + 42$ | Kim et al. (2010) |
| 4-30 | $T = -14 + 55.2 \times TEX_{86}$ | Powers et al. (2010) |
| 10-30 | $T = 3.5 + 38.9 \times TEX_{86}$ | Tierney et al (2010) |
| -2-30 | $T = (TEX_{86} - 0.3038) / 0.0125$ | Shevenell et al. (2011) |
| 14-34 | $T = 32.873 \times \ln(GDGT\ index\text{-}1) + 50.771$ | Hollis et al. (2012) |
| 14-34 | $T = 39.036 \times \ln(TEX_{86}) + 36.455$ | Hollis et al. (2012) |
| 15-35 | $T = (TEX_{86} - 0.21) / 0.015$ | Qin et al. (2015) |
| 10-30 | $TEX_{86} = -0.0006T^2 + 0.023T + 0.33$ | Qin et al. (2015) |
| 10-25 | $TEX_{86} = -0.0017T^2 + 0.054T + 0.11$ | Qin et al. (2015) |
| 2-10 | $T = 27.898(TEX_{86}^{L}) + 22.723$ | Harning et al. (2019) |

| Name | Calculations | Reference |
|---|---|---|
| $TEX_{86}$ | $[GDGT\text{-}2]+[GDGT\text{-}3]+[Cren']/[GDGT\text{-}1]+[GDGT\text{-}2]+[GDGT\text{-}3]+[Cren']$ | Schouten et al. (2002) |
| $TEX_{86}'$ | $[GDGT\text{-}2]+[GDGT\text{-}3]+[Cren']/[GDGT\text{-}1]+[GDGT\text{-}2]+[
[revised manuscript text omitted]

---

## Author Response (AR2)

Editor comments (author responses in red)

We appreciate the in-depth comments and suggestions from the editor to improve the manuscript, from general comments to specific line-item comments. We have taken most of these into account, and detail our changes and responses to other suggestions below.

The term "paleotemperature measurements" comes up frequently. In some contexts I think this is appropriate, for example where it's clear that you're referring to measurements undertaken to derive a paleotemperature estimate. In other cases, for example line 23 in the abstract, my sense is that "paleotemperature estimates" is a more appropriate because the datum of information (e.g. MAT at X location Y thousand years ago) was not measured but rather estimated from measurements.

In the abstract and introduction, we do mean to refer to the measurement (act) rather than estimate (measurement output). However, we recognize that the description did not make that clear. We have modified the text accordingly (lines 23-25, 65-67) to make this more clear.

The point about uncertainty and geological time is an important one and it's elegant to turn to Urey's pioneering work to make the point. But phrasing of the point is a bit awkward and I recommend some wordsmithing.

We agree that the wording was awkward there – we have modified this to improve clarity.

Line 186: You may not agree, but it seems optimistic to suggest that calibration can "effectively remove" the influence of CCFs in even the most well-characterized proxy system. Is it more realistic to instead say "account for" or "mitigate against"?

We agree – while control measurements do "effectively remove" the influence of CCFs, proxies (the subject of the paragraph in question) do, at best, account for CCFs. This has been changed accordingly.

The section from lines 305 to 320 is interesting, and I'm not sure if I agree with the assessment that sample processing, human error, etc give rise to "unquantifiable uncertainty". Tools such as secondary reference standards (ie separate from instrument calibration standards), replicate analyses, and process blanks are all ways to account for and, in some cases, quantify the impacts of sample handling. I suggest a long chat with someone who runs a mass spectrometer for a living… they would likely have some good thoughts on this.

We recognize that the structure and wording in that paragraph was not clear. We have modified that section to better make clear how unquantified and quantified uncertainties relate to analytical uncertainty, making a more straightforward description of how researchers can indeed quantify such uncertainty.

Re: transfer functions, I was surprise to see the discussion come up relatively late. Since transfer functions are the sets of equations used in "correlation constrained proxies", it would seem natural to bring up the term there. This would also provide an opportunity to give some more examples of correlation constrained proxies (e.g. chironomids, pollen, ice core/permafrost water isotopes, etc). This would contribute to addressing, in part, the next point…

We don't quite agree that transfer functions are universally used by correlation-constrained proxies to the point that they should be brought up immediately when introducing those proxies. We see them as more of a tool that is employed to push correlation-constrained proxies closer to the observation/inference-constrained proxy side (by identifying CCFs and their relative influence). Where the discussion of transfer functions is situated currently represents that idea of transfer functions as tools used to improve our understanding of proxies, ultimately to push them away from correlation-constrained proxy.

Reading the manuscript again, there's a perception of beating up a bit on the TEX proxy. I'm not sure if this is intended or not (or maybe I'm reading more into it because I'm also handling a manuscript that is critical of current TEX calibration schemes). I can't point to anything specific, but if that wasn't your intention I'd urge you to take a close read of the sections relating to TEX and see if the tone can be subtly shifted to avoid that perception. You could also consider adding a few sentences of text to summarize some of the major contributions made by workers applying TEX to paleoclimate problems.

We, like many others, do think that too much confidence has been ascribed to the TEX86 proxy, given its relationships with CCFs we outline in our manuscript. In many ways, our manuscript provides an epistemological critique of the proxy that is meant to highlight shortcomings and room for improvement. We have changed the language of some sections (e.g., lines 317-330) to focus more on how the data is represented (rather than how researchers represent the data), and feel that our discussions of the ongoing work to improve TEX leave the paper with an optimistic tone.

Fig 1: the numbers on the X axis seems impossibly low to me. Surely there are (at least!) 90 papers a month published in various journals that discuss paleotemperature. You don't need this figure to support the point that there has been an increasing scholarly interest in paleoclimate over the last several decades, so I suggest deleting it or re-doing the analysis with more inclusive search criteria.

We agree that the figure adds very little to the manuscript, and have removed it.

Re: "first quantitative paleotemperature proxy" (line 152), we are not aware of any truly quantitative paleothermometer prior to 1950 – neither tree rings nor varved sediments, to our knowledge, provided/provide quantitative temperature estimates. If the editor is aware of such applications, we are happy to change this language, but currently feel that the d18Ocalcite proxy is well known to be the first.

We have also made small edits throughout the manuscript to improve clarity elsewhere.